# Computational and Experimental Analysis of Surface Residual Stresses in Polymers via Micro-Milling

**DOI:** 10.3390/polym16020273

**Published:** 2024-01-19

**Authors:** Fuzhong Sun, Guoyu Fu, Dehong Huo

**Affiliations:** 1School of Mechanical and Power Engineering, Nanjing Tech University, Nanjing 211816, China; sunfuz@njtech.edu.cn; 2Key Laboratory of Mechanism Theory and Equipment Design of Ministry of Education, Tianjin University, Tianjin 300072, China; 3Tianjin Key Laboratory of High Speed Cutting and Precision Machining (TUTE), Tianjin 300222, China; 4School of Engineering, Newcastle University, Newcastle upon Tyne NE1 7RU, UK; dehong.huo@newcastle.ac.uk

**Keywords:** polymer, surface residual stresses, Mulliken–Boyce model, milling process

## Abstract

This research conducts an in-depth investigation into the residual stresses in resin micro-milling processes. Considering that resin is the most crucial matrix material in composites, the construction of a precise machining theory for it is not only key to achieving high-quality- and efficient processing of composite materials but also fundamental to enhancing the overall performance of the materials. This paper meticulously examines the surface integrity and accuracy of epoxy polymers following precision machining, primarily revealing the significance of residual stresses and size effects in extending the lifespan of precision components and promoting their miniaturization. We have adopted an innovative finite element (FE) simulation method, integrated with the Mulliken–Boyce constitutive model, to profoundly analyze the impacts of residual stresses on the surfaces and sub-surfaces of thermosetting polymers. This research further explores the influence of critical machining parameters such as chip thickness, cutting edge radius, feed per tooth, and axial depth on cutting forces, as well as the inherent size effects in polymers. Utilizing X-ray diffraction (XRD) technology, we accurately measured the residual stresses generated during the micro-milling process. The close correlation between FE simulations and experimental results validates the accuracy and effectiveness of our method. This study represents a substantial breakthrough in finite element simulation techniques for high-precision machining of polymer materials, injecting valuable theoretical and practical knowledge into the field.

## 1. Introduction

Mechanical and thermal interactions between the tool and workpiece during machining induce surface residual stresses while preserving the internal structure’s integrity. These stresses, with their uneven distribution, profoundly affect the machined surface quality and impose complexities on subsequent manufacturing processes [1]. In response to the burgeoning demand for rapid and precise micro-product fabrication, micromachining techniques, particularly micro-milling, have attracted significant industrial attention. Distinguished by its capacity to fabricate micro-runners and intricate surfaces, micro-milling stands out as a critical technique, surpassing the scope of conventional micromachining methods. This study focuses on micro-milling, endeavoring to explore and refine innovative machining strategies at both the micron and nanometer scales. The aim of this study is to broaden the scope of micro-milling applications within the domains of precision and ultra-precision machining [2,3], thereby advancing micro-machining technology and understanding its extensive implications across diverse sectors.

Thermosetting polymers, exemplified by epoxy resins, find extensive applications in precision manufacturing. With the dramatic surge in demand for polymeric materials across industries, there is an imperative for an in-depth study of their processability. Such an exploration will facilitate a profound understanding of their machinability, unveiling novel processing techniques [4,5]. In pursuit of innovative methods and emerging application domains for materials akin to epoxy, numerous studies have embarked on investigating the machinability of thermoset polymers. For instance, Wan et al. [6] inferred that stress concentration, during the cutting process, detrimentally affects the mechanical properties of thermoset polymers, attributing this to the presence of porosity and notches. Similarly, Xiao and Zhang [7] highlighted that the viscous deformation of polymers plays a pivotal role in determining the quality of machined surfaces, thereby broadening comprehension of thermoset polymers and their machinability.

Nevertheless, existing research into residual stresses on thermoset plastic surfaces is limited in depth [8,9]. Various machining processes, including milling and turning, are known to induce significant residual stresses on the surfaces of machined components. Such stresses critically undermine the component’s durability, contributing to fatigue, creep, deformation, and increased vulnerability to stress corrosion. These residual stresses originate from uneven plastic deformation, distinct thermal gradients, and phase transformations occurring during machining. Notably, plastic deformation arises when the applied stress exceeds the material’s yield strength. On machined surfaces, plastic strains are notably more pronounced than on sub-surfaces, a result of the variation in strain hardening rates between the material’s surface and its core. This variance leads to the entrapment of stress within the workpiece, which remains even after external load removal [10]. Therefore, to achieve ultra-precision in machining thermoset polymers, a detailed understanding of their surface residual stress patterns is crucial.

Although experimental and finite element models for predicting residual stresses are prevalent in scholarly texts, it is notable that a majority of these investigations predominantly focus on metallic materials [1,11,12], leaving the study of polymers considerably underrepresented. This research aims to fill this critical gap by introducing an innovative Mulliken–Boyce model specifically designed to simulate residual stresses in polymeric substances. Additionally, addressing the prevalent inclination towards 2D orthogonal cutting in the existing literature, this study introduces a ground-breaking 3D cutting model simulation tailored for micro-end milling. This novel approach integrates both macro and micro-scale machining processes. Significantly, this research marks a departure from the traditionally dominant molecular dynamics (MD) simulation approach [13,14], heralding a fundamental shift in the simulation methodologies prevalent in this field.

Calamaz et al. [15] developed a model based on two-dimensional finite element analysis, primarily elucidating chip formation and shear deformation during titanium alloy machining. This model predicts the impact of mechanical strain on titanium alloy surfaces during machining, considering strain, strain rate, and thermal softening in the contact zone. Similarly, Liu et al. [16] empirically delineated the distribution of residual stresses from mechanical interactions on machined surfaces, comparing stress outcomes from orthogonal, diagonal, and conventional cuts to ascertain the influence of cut depth on subsurface residual stresses. Pan et al. [17] utilized the finite element method to simulate residual stresses in Ti-6Al-4V, while Davoudinejad et al. [18] developed a 3D finite element model for micro-end milling of aluminum, examining tool runout’s effects on chip evacuation and cutting forces. These studies collectively highlight the necessity for robust models to accurately predict and analyze the complexities of residual stresses and their effects on machined materials.

A diverse array of simulation techniques has emerged for the modeling of polymer materials, including analytical [19], mechanical [20], and finite element analysis (FEA) [21,22]. Of these, FEA is frequently lauded as the most effective, offering simulations that more accurately reflect the real-life processing of polymers [23,24]. Theoretically, molecular dynamics (MD) methods are better suited for the intricacies of polymer machining, as MD models meticulously illustrate atomic and molecular interactions during deformation [25], providing a detailed view of stress responses throughout the entire cutting process. However, the development of FEA has significantly reduced the duration of simulations, shifting from extended periods on advanced workstations to just hours on standard laptops, while producing results that more accurately replicate actual three-dimensional machining scenarios [25]. This efficiency has spurred a shift from MD to FEA in the realm of polymer material processing simulations.

In short, this study will delve into the issue of residual stresses encountered in the micro-milling of resin materials. Considering the critical role of resin as a matrix within composite materials, constructing a theory of its precision machining is not only key to achieving high-quality- and efficient processing of composites but also fundamental to enhancing the overall performance of the materials. This paper conducts a detailed investigation into the surface integrity and accuracy of epoxy polymer materials following precision machining, emphasizing the significant role of residual stresses and size effects in extending the lifespan of precision components and promoting their miniaturization. Additionally, a novel finite element cutting model tailored for thermosetting polymers, such as epoxy resins, is introduced. Developed using Abaqus 2020 software, this model is dedicated to simulating the micro-milling process. The goal of this research is to explore the relatively uncharted territory of ultra-precision cutting of polymeric materials, aiming to fill the existing gaps in knowledge. This innovative analysis of surface residual stresses deciphers the complex mechanisms behind the precision machining of thermosetting polymers. Involving extensive experimental and simulation studies, it provides in-depth insights into cutting forces, chip formation mechanisms, strain localization, and stress distribution, thereby significantly expanding the current corpus of knowledge.

## 2. Polymer Constitutive Model

### 2.1. Polymer Behavior

Figure 1, offering a comparative visual analysis of the internal mechanical behaviors of polymers versus metals during machining, not only clarifies the distinct molecular-level differences under stress but also underscores the unique challenges in polymer processing. Figure 1a illustrates the rotation of molecules within a polymer material during deformation, while Figure 1b offers a one-dimensional rheological interpretation of the Mulliken–Boyce constitutive model. The notable differences in ductility between polymers and metals largely arise from the inherent impurities and voids within the polymer matrix. Therefore, the development of constitutive material models for polymers is essential to elucidate the impact of these polymers on material properties at the microscopic level. Several constitutive models for polymer materials have been proposed, including, but not limited to, the Arruda–Boyce [26] and the Mulliken–Boyce [27] models. However, the integration of these models into commercial Finite Element (FE) commercial software is an ongoing process.

For the exploration of polymer machining characteristics using commercial FE software, the Mulliken–Boyce model was selected as the computational framework for this analysis. The implementation of this model was executed via the VUMAT subroutine, programmed in the Fortran language. Grounded in the principles of large deformation plasticity, the Mulliken–Boyce model is a physically based constitutive model that effectively characterizes the elastic, yield, and post-yield behavior of polymers across various strain rates. This model incorporates two active molecular mechanisms, each with distinct elasticity and viscoelasticity properties. Operating concurrently, these mechanisms include a non-linear entropy-hardening component [28], providing a comprehensive understanding of the material’s behavior. Concurrently, the methodology for scripting the Mulliken–Boyce constitutive model in VUMAT has been previously published by the authors of this article [25,29,30].

### 2.2. Mulliken–Boyce Model Parameters for Epoxy

A genetic algorithm was employed to assess the experimental data using a one-dimensional version of the Mulliken–Boyce model. This algorithm helped elucidate the constitutive parameters that dictate the stress–strain response of epoxy resins, as detailed in Table 1 [31]. To integrate the constitutive model into this research, the vectorized user material (VUMAT) method was utilized. The model’s validity was ascertained by juxtaposing the modeling outcomes in uniaxial compression against previously published experimental findings. Validation experiments carried out by Fu et al. [30,32] further substantiated the accuracy of the model.

## 3. FE Machining Model Setup

The foundational components of this study are depicted in Figure 2. Figure 2a showcases the setup of the 3D finite element (FE) cutting model used in micromachining, while Figure 2b presents the geometry and meshing associated with the 3D FE cutting model. The model features a stationary workpiece and a rigid tool as the primary subjects of analysis, with the tool advancing horizontally at a predetermined speed. For the epoxy resin examined in this research, a ductile damage model was employed, encompassing fracture strains, triaxial stresses, and strain rates of 0.04042, 7.815, and 0.02556, respectively [33,34,35]. Utilizing the ductile material damage model in Abaqus, the polymer material undergoes linear degradation once the change in equivalent effect surpasses a critical point [36].

In the cutting process, post-yield deformation may initiate additional dissipation mechanisms such as damage and molecular rearrangement. Nevertheless, owing to their elusive nature, these mechanisms are not directly addressed in this study. Consistent with the observations of Varghese and Batra [37] and Sun and Gamstedt [38], the hypothesis is that all plastic work transforms into heat, devoid of thermal diffusion, thereby establishing an adiabatic heating mechanism. The ambient temperature during the cutting operation is maintained at 25.0 °C. The model comprises 1,120,172 elements and 1,115,751 nodes. The selected element type, C3D8R, is an 8-node linear brick with reduced integration and hourglass control, chosen for its proven stability and efficiency in handling complex geometries and material behaviors [39].

For scenarios involving intricate stress states and geometric features, consideration of higher-order elements such as C3D10, a 10-node tetrahedral element, and C3D20R, a 20-node quadratic brick element with reduced integration, is recommended. The C3D10 element provides adaptable meshing capabilities suitable for complex shapes, while C3D20R is preferred for its ability to capture detailed stress concentrations. Employing these elements is anticipated to significantly enhance the model’s precision and reliability, particularly in scenarios characterized by high stress gradients and intricate boundary conditions [39].

The model serves to investigate the impact of different cutting radii on chip deformation [40]. In this simulation, cutting-edge radii of 1.0 µm, 1.5 µm, 2.0 µm, 4.0 µm, and 6.0 µm were examined. Moreover, the undeformed chip thickness is set at seven levels: 0.5 µm, 1.0 µm, 1.5 µm, 2.0 µm, 4.0 µm, 6.0 µm, and 8.0 µm. The tool’s leading and clearance angles are set at 30.0° and 6.0°, respectively. The model’s dimensions for length, width, and height are 40.0 µm, 12.0 µm, and 18.0 µm, respectively. The simulation parameters, including the tool angle, align with those used in the confirmatory experiments conducted in this research. Table 2 provides an exhaustive list of the cutting model parameters and conditions applied in the FE cutting model.

In the field of machining, the undeformed chip thickness influences surface quality. Increasing the cutting radius typically enhances surface smoothness by optimizing the distribution of forces and heat, thereby reducing surface roughness. However, this improvement is not without limits; excessive increases in radius may induce vibrations and deflection, compromising part precision. Additionally, material properties and tool wear patterns also impact the selection of the optimal cutting radius. Therefore, determining the cutting radius requires a comprehensive consideration of material response, tool durability, and machining outcomes. Employing advanced computational methods, such as finite element analysis, can enhance the accuracy and efficiency of this selection process. In the pursuit of precision machining, engineers and researchers must meticulously balance various factors to achieve optimal processing quality and efficiency.

## 4. Micro-Milling Experimental Setup

The experimental procedures were executed using an ultra-precision tabletop machine tool, the Nanowave MTS5R. This device operates with a steady power output of 100 W (240 V) and can reach a maximum spindle speed of 80,000 rpm in dry conditions. It features three axes (X, Y, Z), each with a resolution of 0.1 μm, and is driven by a DC servo motor. To measure the cutting forces, a Kistler cutting force dynamometer (9256C2) was utilized, as depicted in Figure 3a. Figure 3b–d provide schematic representations of the 3D micro-milling and orthogonal cutting processes. 

In the micro-drilling process, the inherent mass of the drill bit and its high-speed rotation generate inertial effects. These effects can lead to the drill bit exceeding the predetermined cutting path when stopping or changing direction, thus adversely affecting the precision and quality of the hole and subsequently the Material Removal Rate (MRR). This study identified optimal operational parameters, such as cutting speed and feed rate, to mitigate the impact of inertia while maintaining high MRR. Furthermore, the milling machine was positioned on a vibration-damping table to absorb the vibrations and shocks induced by inertia, thus minimizing disturbances to the cutting trajectory.

Lei et al. [41] demonstrated that the maximum uncut chip thickness (approximately 100 μm) is notably smaller than the tool diameter (1000 μm). This results in a minor variation in uncut chip thickness compared to the distance covered during a 180° tool rotation (around 1570 μm). Consequently, the micro-milling feed rate aligns with the undeformed chip thickness observed in orthogonal machining. Table 3 presents the specifications of the uncoated 1 mm diameter end-mill tool used during the machining process. The X-ray diffraction (XRD) technique was employed to evaluate the residual stresses on the surface and deeper layers after micro-milling. The Sin2Ψ method was implemented, using an X-ray tube derived from a copper (Kα) target.

Table 4 provides a detailed account of the micro-milling conditions for this experimental segment. The depth of cut was established at 100 µm, with the cutting and spindle speeds set at 31.4 m/min and 10,000 rpm, respectively. Feed per tooth (FPT) values ranged from 0.5, 1.0, 1.5, 2.0, 4.0, 6.0, to 8.0 µm/tooth. As detailed in the preceding section on the cutting simplification process, the FPT corresponds to the undeformed chip thickness in the FE simulations.

## 5. Results and Discussion

The size effect phenomenon is a unique characteristic of micromachining and has garnered significant attention in the realms of precision and ultra-precision machining research [42,43,44]. During the micro-milling of materials, chip formation may not be consistent with each tool pass when the feed per tooth (FPT) is minimal. As the feed per tooth approaches the cutting-edge radius of the tool, it becomes a decisive factor for the transition from shear to ploughing mechanisms [45,46].

Materials that undergo significant ploughing or friction primarily manifest as ploughed matter, accompanied by subsequent elastic rebound with each tool pass, leading to the absence of chip formation during each tool trajectory [47,48]. When the thickness of uncut chips falls below a certain critical threshold, size effects can adversely influence the cutting process, resulting in amplified cutting forces, reduced tool lifespan, and a decline in surface dimensional quality. Therefore, the ploughing region directly impacts the surface texture of the material, intensifying the cutting forces. A comprehensive understanding of these dimensional effects is crucial for achieving precision and ultra-precision machining in thermoset polymer materials [49].

Figure 4a presents a graphical depiction of the chip formation dynamics throughout the finite element (FE) cutting process. It illustrates the gradual increase in undeformed chip thickness from an initial zero to its maximum, aligning with the feed per tooth. This visual representation sequentially highlights the various stages the cutting tool experiences while interacting with the workpiece. Initially, as the tool begins engagement with the workpiece, the undeformed chip thickness is minimal, leading to a predominance of rubbing against the workpiece due to inadequate penetration.

In contrast, Figure 4b delineates the FE cutting process focusing on material removal under the specified cutting parameters: a 2.0 µm undeformed chip thickness and a 1.5 µm cutting tool radius. Notably, the initial phase of tool–workpiece interaction is characterized by observable lateral material flow, indicating a prevalent topological manipulation or ploughing action. The cessation of lateral material flow signifies a critical transition from ploughing to a shear mechanism, corresponding to the minimum undeformed chip thickness. Furthermore, the von Mises stress measurements employed in this study are quantified in megapascals (MPa).

Figure 5 illustrates the stress distribution and the physical deformation during the micro-milling process of a polymer workpiece. Figure 5a illustrates a side perspective of the workpiece post-machining, accentuating the chip formation that results from the cutting process. This depiction is critical for understanding the morphological characteristics of the chips, which are indicative of the prevailing cutting mechanics. In Figure 5b, a topographical view of the machined surface is rendered, correlating to specific machining parameters: an undeformed chip thickness of 2.0 µm and a cutting tool radius of 1.5 µm. The progressive advancement of the tool is marked by an increased undeformed chip thickness, which primarily facilitates material removal via ploughing or extrusion.

Initially, the material is predominantly displaced through plastic deformation during the frictional and ploughing stages, rather than being segmented into particulate debris. This stage of deformation prompts a lateral material flow, which is a crucial aspect to consider when analyzing the integrity of the machined surface. As the machining operation advances, an escalation in the uncut chip thickness is observed, leading to a transition towards a shearing-dominant removal mechanism. This evolution diminishes the relevance of frictional and ploughing activities in the material removal process [50]. Notably, when the material removal is chiefly governed by shearing actions, the absence of lateral flow becomes a characteristic feature. The chips generated under these conditions, as depicted in Figure 5, are serrated in morphology. This serrated appearance is a manifestation of the high-speed shearing dynamics involved in the machining of the materials. It contrasts with continuous chip formation and is a critical indicator of the material’s behavior under the influence of the defined cutting conditions.

Figure 6 provides a comprehensive Finite Element (FE) model depiction of the cutting mechanism, critically analyzing the effects of varying the cutting-edge radius with a constant undeformed chip thickness of 2.0 µm. This analysis focuses on the consequent changes in chip formation and cutting force dynamics, as referenced in [50,51]. The FE simulation identifies a critical chip thickness of approximately 1.0 µm, which represents 67% of the 1.5 µm cutting-edge radius. This finding underscores the sensitivity of chip formation to the tool’s geometrical features. Subsequently, Figure 7 demonstrates the feed and thrust forces as simulated in the FE model, operating under the specified conditions of 2.0 µm undeformed chip thickness and a 1.5 µm cutting tool radius. In a related context, Figure 8 delineates the resultant feed and thrust forces from the FE simulation, maintaining a consistent undeformed chip thickness of 2.0 µm while varying the cutting-edge radii. The graphical data reveal an escalation in FE cutting forces proportional to the increase in the cutting-edge radius. The collective insights from the FE analysis suggest a direct correlation between the tool edge radius and the magnitude of cutting forces and residual stresses. This phenomenon can be attributed to the dimensional constraints that facilitate material extrusion between the tool and the workpiece. Such an understanding is pivotal in optimizing tool design and machining parameters to enhance efficiency and ensure the integrity of the machined surface.

Figure 9 presents a finite element (FE) simulation of the cutting operation, focusing on the force dynamics involved in material removal. The simulation is conducted under specific conditions: an undeformed chip thickness of 2.0 µm and a range of cutting tool radii. A variation in undeformed chip thickness is observed to significantly influence the stress distribution within the polymer material, which is notably sensitive to stress changes. This sensitivity leads to a transition from ductile to brittle states in the polymer as the undeformed chip thickness increases. Consequently, this shift results in the formation of varying chip morphologies, including continuous, serrated, and discontinuous chips, directly sheared from the matrix material. Further analysis of chip deformation across different tool radii reveals that larger radii tend to produce more fragmented chips. This fragmentation is attributed to the increased propensity of larger radii to initiate the primary shear direction deeper within the workpiece. With the enlargement of the tool radius, there is a gradual shift in the shear direction towards a positive shear angle, culminating in the formation of a shear slip surface. The chip-cutting process is characterized by intermittent dynamics due to the localized concentration of shear within the material.

These observations corroborate the hypothesis that the deformation mechanism of epoxy chips can be predominantly classified as shear plastic slip. This conclusion lends substantial credence to the findings of previous empirical studies [50] and FE simulations [25]. In a complementary manner, Figure 10 illustrates the simulation results concerning shear stresses under the set conditions of a 2.0 µm undeformed chip thickness and a 1.5 µm cutting tool radius.

Figure 11a displays the finite element (FE) simulation of the resultant force and material removal under the cutting parameters of an undeformed chip thickness of 2.0 µm and various cutting tool radii. As the undeformed chip thickness increases, the FE cutting force also rises. Figure 10b compares the resultant force obtained from the FE simulation with the experimental data. In the experiment, the cutting forces initially showed a decrease, followed by an increase, which might be due to size effects. However, this trend was not observed in the FE results.

X-ray Diffraction (XRD) was utilized to ascertain the values of residual stresses. Figure 12 delineates a comparative analysis of residual stresses ascertained from Finite Element (FE) simulations and experimental observations, under specified cutting parameters: an undeformed chip thickness of 2.0 µm and a cutting tool radius of 1.5 µm, considering both the cutting and transverse directions. The FE analysis identified the most pronounced residual stresses, situated 2.5 μm beneath the machined surface, in the cutting (feed) and transverse (orthogonal to the feed) directions, with compressive stress values recorded at 36.4 MPa and 24.8 MPa, respectively. These stresses were predominantly compressive in nature. Correspondingly, experimental findings confirmed that the most intense residual stresses in both the cutting and transverse directions, positioned 2.5 μm below the machined surface, are compressive, with values recorded at 32.1 MPa and 20.3 MPa, respectively. These data exhibit a high level of congruence with the FE simulation outcomes, underscoring the reliability and accuracy of the simulation process in predicting residual stress patterns.

The peak of residual stresses is observed at a certain depth from the machined surface. Beneath the machined surface, the material exhibits increased resistance due to variations in strain gradients at different depths, which can be attributed to the accumulation of high-density dislocations. However, the magnitude of the residual stress decreases at greater depths, eventually reaching zero at a depth of 16.3 µm from the machined surface.

## 6. Conclusions

This research developed a sophisticated finite element (FE) micro-cutting model specifically for polymer materials, utilizing the advanced Mulliken–Boyce model. This study implemented finite element simulations to investigate the size effects and inherent residual stresses within polymers comprehensively. Notably, the results derived from the FE model demonstrate a high degree of correlation with empirical observations, particularly in terms of the trends and magnitudes of the residual stresses observed. The adoption of a three-dimensional finite element approach provides a robust and detailed framework for understanding strain localization phenomena during the cutting of thermosetting polymers, such as epoxy. This study’s key contributions include the following:

The primary deformation mechanism in epoxy cutting is identified as shear-plastic slip, aligning with prior experiments showing continuous, serrated, and fragmented chips. The simulation’s agreement with theoretical and empirical research affirms the methodology’s validity.

Finite element analysis indicates increasing cutting forces and residual stresses with larger tool edge radii, primarily due to size effects leading to material extrusion, significantly impacting the cutting process.

The most pronounced residual stresses, observed approximately 2.5 μm below the machined surface in both cutting and orthogonal directions, are quantified at 36.4 MPa and 24.8 MPa, respectively, pinpointing critical stress concentrations.

Experimental data, showing peak compressive residual stresses at 32.1 MPa and 20.3 MPa 2.5 μm below the surface, corroborate the FE model, validating the simulation’s reliability and applicability.

## Figures and Tables

**Figure 1 polymers-16-00273-f001:**
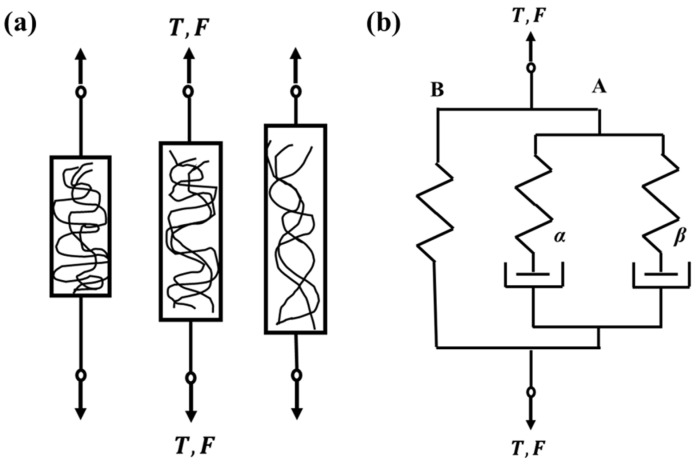
(**a**) Illustration of molecular rotation within a polymer material during deformation; (**b**) unidimensional rheological representation of the proposed constitutive model [28].

**Figure 2 polymers-16-00273-f002:**
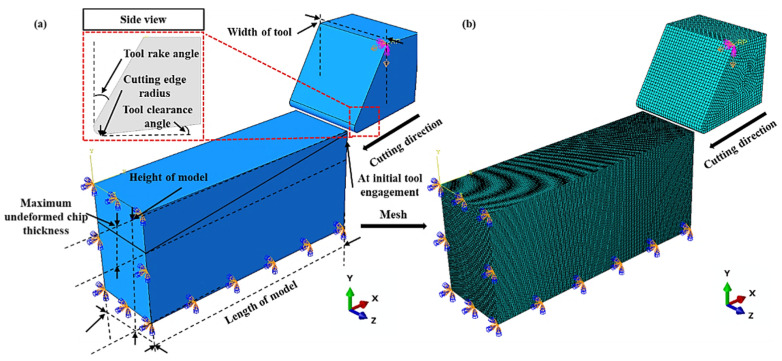
(**a**) 3D FE cutting model setup for micromachining; (**b**) geometry of the 3D FE cutting model with mesh.

**Figure 3 polymers-16-00273-f003:**
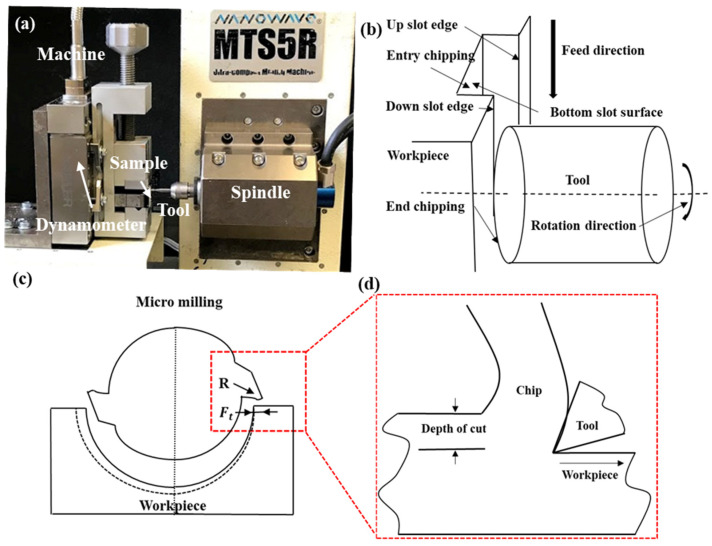
(**a**) Depiction of the experimental setup; (**b**) conceptual diagram of the 3D micro-milling process; (**c**) schematic representation of the 2D micro-milling operation, with (**d**) illustrating its correlation with the orthogonal cutting process.

**Figure 4 polymers-16-00273-f004:**
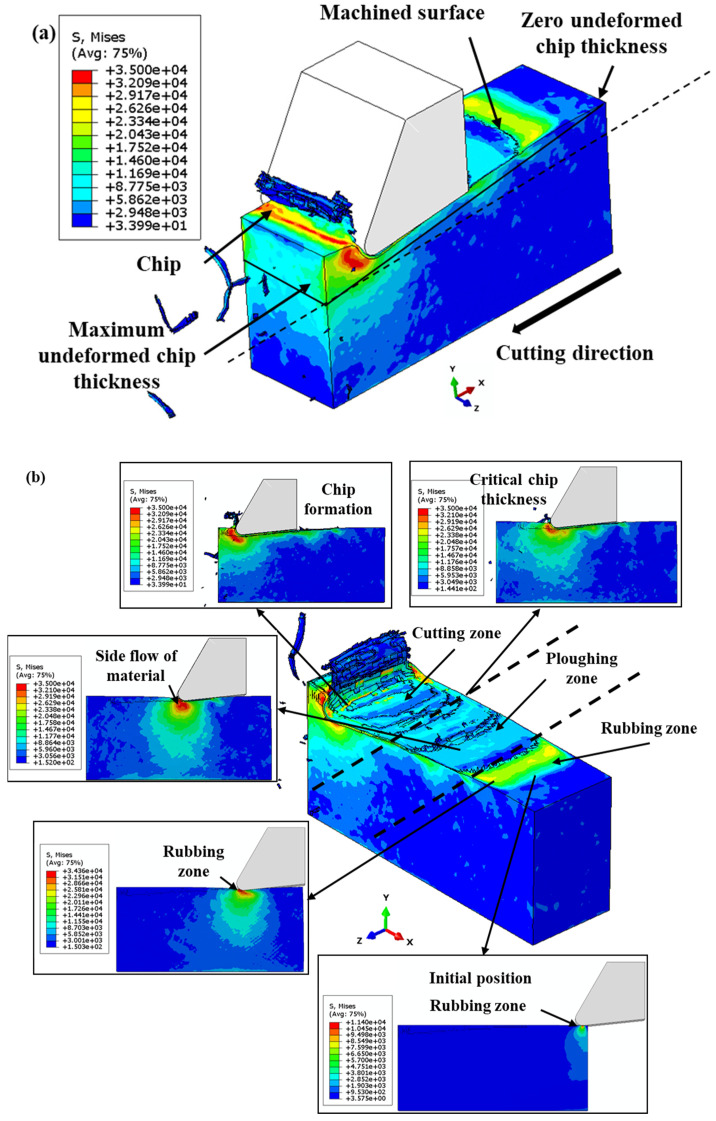
(**a**) Illustration of the chip generation sequence in the context of the FE cutting mechanism; (**b**) FE modeling of the cutting methodology, emphasizing material displacement under the cutting specifications of a 2.0 µm undeformed chip thickness and a 1.5 µm cutting tool radius.

**Figure 5 polymers-16-00273-f005:**
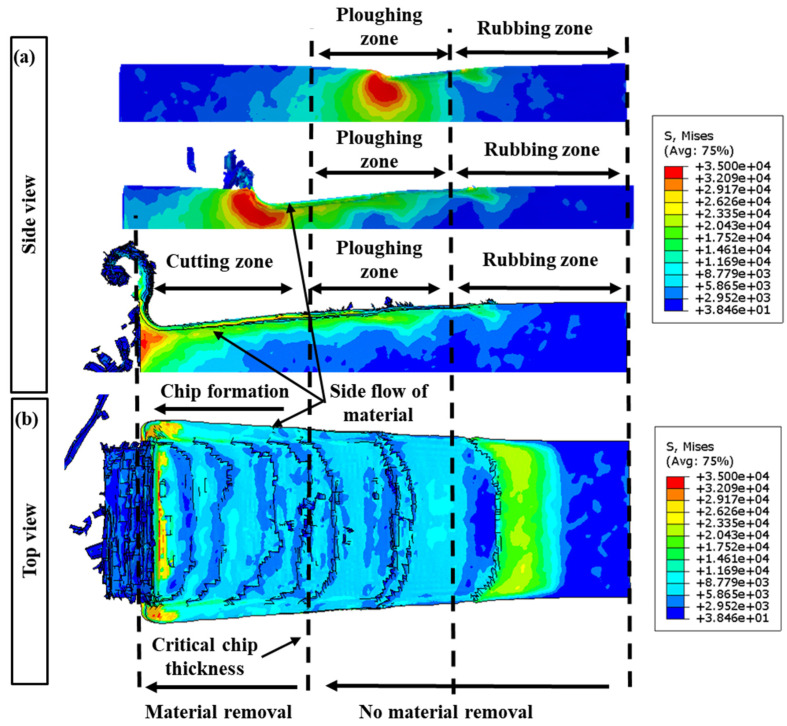
(**a**) Side view of the machined workpiece and chip formation; (**b**) top view of the machined surface at the cutting conditions of 2.0 µm undeformed chip thickness and 1.5 µm cutting tool radius.

**Figure 6 polymers-16-00273-f006:**
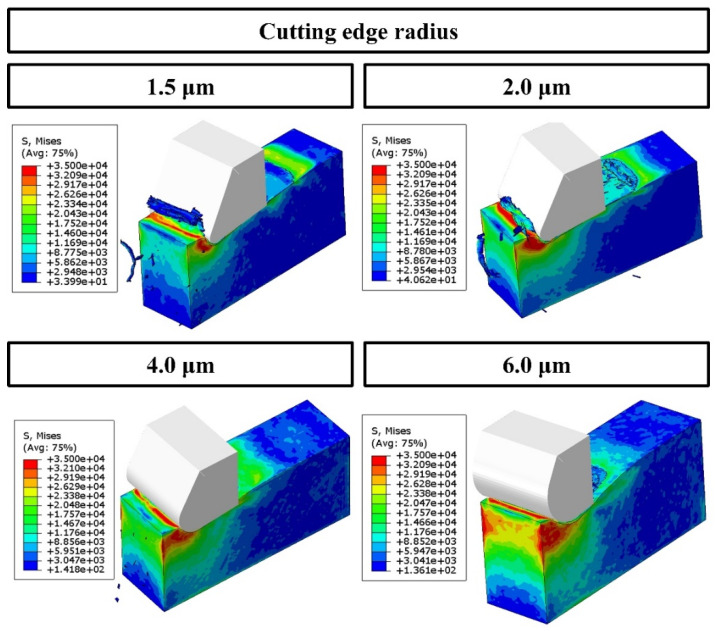
FE simulation cutting process in terms of different values of cutting-edge radius at the cutting condition of 2.0 µm undeformed chip thickness.

**Figure 7 polymers-16-00273-f007:**
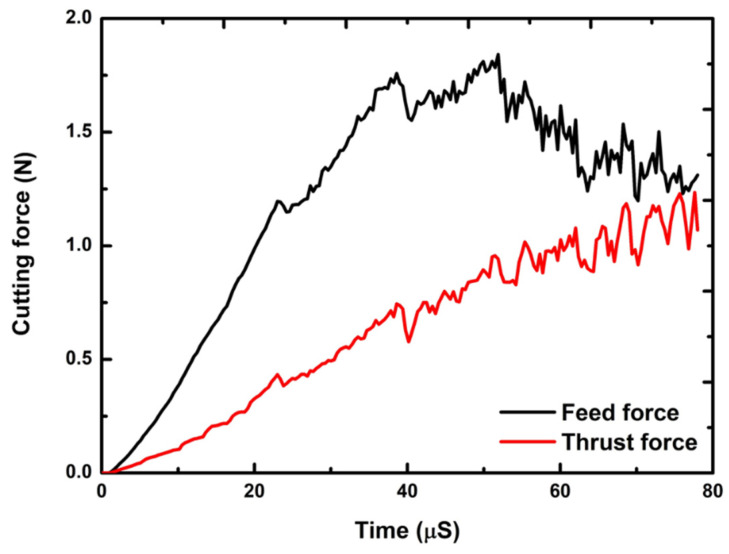
FE simulation feed and thrust forces at the cutting conditions of 2.0 µm undeformed chip thickness and 1.5 µm cutting tool radius.

**Figure 8 polymers-16-00273-f008:**
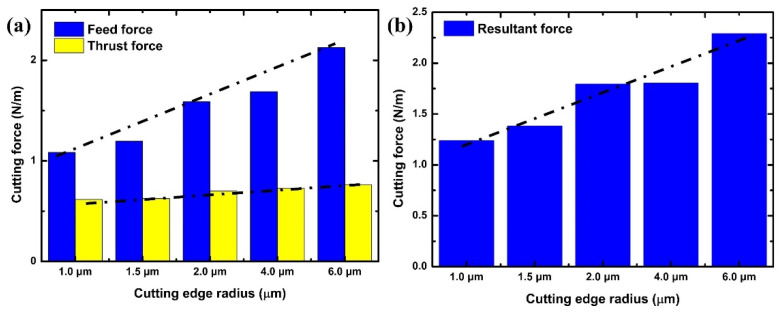
(**a**) Feed and thrust forces as per the FE simulation; (**b**) resultant force under conditions of 2.0 µm undeformed chip thickness and different values of cutting-edge radius simulated in the FE model.

**Figure 9 polymers-16-00273-f009:**
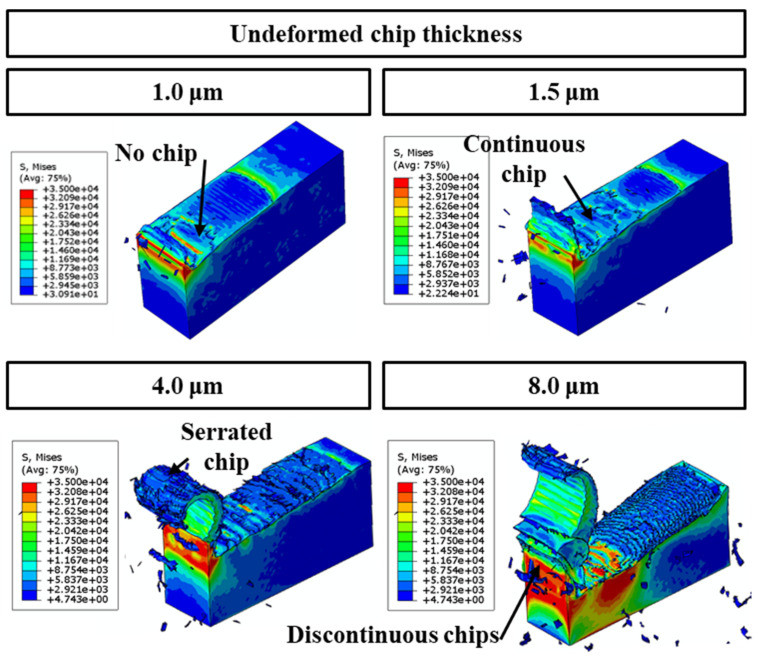
Finite element simulation of the material removal process during cutting, conducted under the conditions of 2.0 µm undeformed chip thickness and various cutting tool radii.

**Figure 10 polymers-16-00273-f010:**
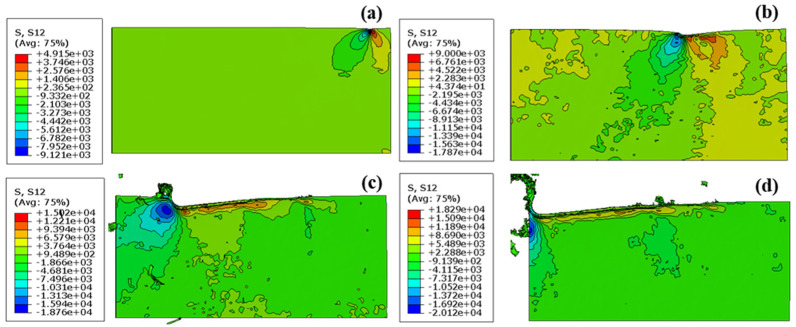
The results of the simulations in terms of shear stresses under the conditions of 2.0 µm undeformed chip thickness and 1.5 µm cutting tool radius. (**a**) at the beginning of cutting process, (**b**,**c**) at the middle of cutting process, (**d**) at the ending of cutting process.

**Figure 11 polymers-16-00273-f011:**
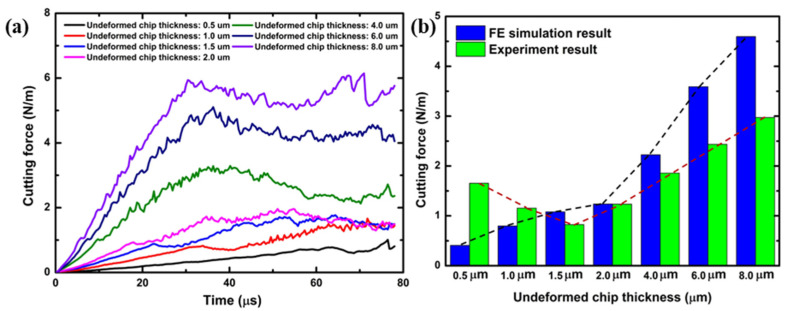
(**a**) FE simulation resultant force; (**b**) comparison of resultant force from FE simulation and experimental results at the cutting conditions of 2.0 µm undeformed chip thickness and different values of cutting tool radius.

**Figure 12 polymers-16-00273-f012:**
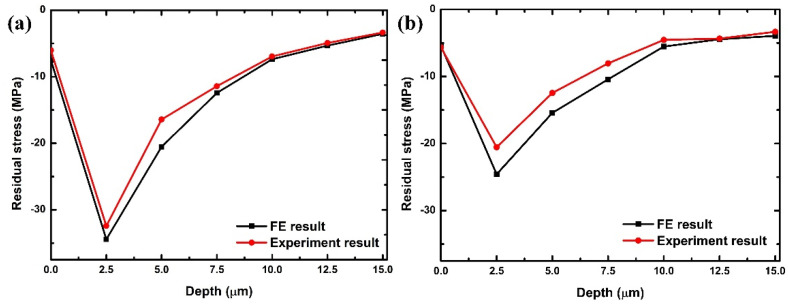
Correlation of finite element simulated and experimental outcomes for resultant force under cutting conditions of 2.0 µm undeformed chip thickness and 1.5 µm cutting tool radius in (**a**) cutting direction; (**b**) transverse direction.

**Table 1 polymers-16-00273-t001:** Epoxy material parameters [31].

Symbol	Units	Value
$S_{ss,a}$	MPa	0.670
${\dot{\gamma }}_{0,a}$	$10^{15}\;s^{-1}$	178.0
${\dot{\gamma }}_{0,\beta }$	$10^{6}\;s^{-1}$	0.542
$\Delta G_{\alpha }$	$10^{-18}\;J$	0.967
$\Delta G_{\beta }$	$10^{-21}\;J$	834
$\alpha_{p,\alpha }$		0.224
$\alpha_{p,\beta }$		0.378
$h_{\alpha }$	MPa	263.0
$C_{R}$	MPa	13.9
N	$m^{-\frac{1}{2}}$	2.02
$\sigma_{\beta }$	MPa	0.283

**Table 2 polymers-16-00273-t002:** Cutting model and condition parameters used in the FE cutting model.

Parameters	Value
Cutting speeds	31.4 m/min
Undeformed chip thickness	0.5 µm, 1.0 µm, 1.5 µm, 2.0 µm, 4.0 µm, 6.0 µm, 8.0 µm
Cutting edge radius	1.0 µm, 1.5 µm, 2.0 µm, 4.0 µm, 6.0 µm
Tool rake angle	30.0°
Tool clearance angle	6.0°
Width of tool	15.0 µm
Width of model	12.0 µm
Length of model	40.0 µm
Height of model	18.0 µm

**Table 3 polymers-16-00273-t003:** Micro-end milling uncoated tool specifications.

Properties	Value
Tool diameter	1.0 mm
Cutting edge radius	1.5 µm
Number of flutes	2
Flute style	Right-hand spiral/medium helix
Finish/coating	Uncoated
Helix angle	30°

**Table 4 polymers-16-00273-t004:** Micro-milling conditions for the experimental section.

Parameters	Value
Cutting speed	31.4 m/min
Spindle speed	10,000 rpm
Feed per tooth	0.5 µm/tooth, 1.0 µm/tooth, 1.5 µm/tooth, 2.0 µm/tooth, 4.0 µm/tooth, 6.0 µm/tooth, 8.0 µm/tooth
Depth of cut	100 µm

## Data Availability

Data are contained within the article.

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
