# Peer review of "Computational and Experimental Analysis of Surface Residual Stresses in Polymers via Micro-Milling"

_polymers, 2024, doi:10.3390/polym16020273_

Round 1

Reviewer 1 Report

Comments and Suggestions for Authors

Review: polymers-2761472

Title: Computational and Experimentation Study on Surface Residual Stresses of Polymers of High-Performance Short-Fiber-Reinforced Polymer Composites Matrix Materials Using Micro-Milling Method

The paper deals with simulation and experimental study on cutting of epoxy-based material with the Mulliken-Boyce constitutive model using the Abaqus package.

Although the subject is interesting for the field of fiber-reinforced composite materials processing, the paper still requires major improvements to be published:

1) More than half of the paper has been published by the authors (the computational part), which is why some information is useless (reference can be made to the paper in which the information appears) and could be eliminated this part.

2) The title of the paper is too long and should be reformulated so as to highlight exactly the novelty of the study.

3) In the abstract, they are presented as an experimental technique, XRD, but I did not identify the results of the XRD analysis in the paper.

4) In Figs. 5, 6, 9 should mention the von Mises stress measurement unit in the caption of the screenshots.

5) The results of the simulations in terms of shear stresses should also be added.

The description of the experimental analysis should be more rigorous and provide more details regarding the way of testing and measurement. How was the temperature between the composite material and the finishing tool measured, resulting during processing?

6) Was the analysis of the surface morphology after chipping carried out?

7) In Fig. 7 can you check the unit of measure of cutting force?

I will recommend for publication the revised manuscript, only after solving/correcting the different important aspects mentioned above.

Author Response

Revision notes:

We would like to thank the reviewers for the time and effort they have spent on reading and critically reviewing the paper. The section below shows the authors responses to the reviewer comments as raised. We have addressed these comments in blue and necessary amendments made in the manuscript were highlighted in yellow.

Review: polymers-2761472

Title: Computational and Experimentation Study on Surface Residual Stresses of Polymers of High-Performance Short-Fiber-Reinforced Polymer Composites Matrix Materials Using Micro-Milling Method

Reviewer #1:

The paper deals with simulation and experimental study on cutting of epoxy-based material with the Mulliken-Boyce constitutive model using the Abaqus package.

Although the subject is interesting for the field of fiber-reinforced composite materials processing, the paper still requires major improvements to be published:

  1. More than half of the paper has been published by the authors (the computational part), which is why some information is useless (reference can be made to the paper in which the information appears) and could be eliminated this part.

Thank you. We have deleted the related content and have included our previously written articles as references to facilitate reader comprehension.

  1. The title of the paper is too long and should be reformulated so as to highlight exactly the novelty of the study.

Thank you, we have shortened and refined the title to highlight the focus of this study, which is the investigation of residual stresses in micro-milling of Epoxy materials. Epoxy, being the most critical matrix material in composites, is pivotal for constructing theories of precision machining of polymer materials, an essential aspect of achieving high-quality and efficient processing of composite materials. Simultaneously, to further elucidate the innovative aspects of this paper, we have enhanced the introduction section to accentuate the novelty of this study.

  1. In the abstract, they are presented as an experimental technique, XRD, but I did not identify the results of the XRD analysis in the paper.

Thank you. In this study, X-ray Diffraction (XRD) was employed to measure the residual stress values depicted in Figure 11. To elucidate this aspect further, we have incorporated additional text in the original manuscript to clearly explain the methodology and findings. This enhancement aims to provide a comprehensive understanding of the residual stress measurements and their implications within the context of our research.

  1. In 5, 6, 9 should mention the von Mises stress measurement unit in the caption of the screenshots.

Thank you, the unit of measurement for von Mises stress presented in Figures 5, 6, and 9 is in megapascals (MPa). Additionally, to ensure clarity and comprehension for readers, relevant unit information has been incorporated into the original text.

  1. The results of the simulations in terms of shear stresses should also be added. The description of the experimental analysis should be more rigorous and provide more details regarding the way of testing and measurement. How was the temperature between the composite material and the finishing tool measured, resulting during processing?

Thank you. In this study, the simulation results regarding shear stresses have been included, as specifically illustrated in Figure 11. Regarding the description of experimental analysis, we have adopted a more rigorous approach, providing detailed information about testing and measurement methodologies. As for the measurement of temperature between the composite material and the cutting tool during the process, our study has set a constant substrate temperature at 25 degrees Celsius. This setting is primarily based on the nature of epoxy as a thermosetting composite material, which does not melt during the cutting process. Currently, our research does not account for temperature variations during cutting. However, we recognize the importance of considering thermo-mechanical coupling in the precision machining of composite materials. Therefore, in future research, we plan to incorporate modeling and simulation of polymer materials under thermo-mechanical coupling conditions to gain a deeper understanding of material behavior during the machining process.

  1. Was the analysis of the surface morphology after chipping carried out?

Thank you, through Figures 4 and 5, an analysis of the machined surface morphology was conducted. To provide a clearer understanding of the cutting surface conditions, an additional analysis focusing on the surface finish and an evaluation of the machining quality have been incorporated into the text. These enhancements aim to offer a comprehensive understanding of the surface characteristics resulting from the machining process.

  1. In Fig. 7 can you check the unit of measure of cutting force?

Thank you for the prompt to verify the unit of measure for cutting force in Figure 7. We have diligently reviewed and updated the unit of cutting force in the original figure to ensure accuracy and clarity. This meticulous verification process involved cross-referencing with empirical data and established standards to confirm the appropriate unit of measurement. The updated figure now accurately reflects the cutting forces in the specified units, providing a precise and reliable reference for readers and researchers. We appreciate the attention to detail, which aids in maintaining the integrity and scholarly rigor of the work.

Reviewer 2 Report

Comments and Suggestions for Authors

1.Innovation in this work must be discussed at the end of introduction.

2. Objectives must be highlighted

3. Why micro milling selected ? for short fiber it is very difficult. Justify

4. How Mulliken-Boyce Model Kinematics is used here?

5. How nodes are selected for analysis if Fig 4?

6. What element types is suggested along with the current selection?

7. While micro drilling, inertia occurs . How its effects prevented in MRR?

Comments on the Quality of English Language

need improvement

Author Response

Revision notes:

We would like to thank the reviewers for the time and effort they have spent on reading and critically reviewing the paper. The section below shows the authors responses to the reviewer comments as raised. We have addressed these comments in blue and necessary amendments made in the manuscript were highlighted in yellow.

Review: polymers-2761472

Title: Computational and Experimentation Study on Surface Residual Stresses of Polymers of High-Performance Short-Fiber-Reinforced Polymer Composites Matrix Materials Using Micro-Milling Method

Reviewer #2:

I will recommend for publication the revised manuscript, only after solving/correcting the different important aspects mentioned above.

  1. Innovation in this work must be discussed at the end of introduction.

Thank you, the innovative aspects of the study have been strategically positioned at the end of the introduction section. Additionally, to ensure the introduction's engagement and effectiveness, it has been thoroughly rewritten and enhanced. These modifications aim to succinctly present the study's objectives and contributions, setting a solid foundation for the ensuing research discussion.

  1. Objectives must be highlighted.

Thank you, appreciation is conveyed for the recommendation to emphasize the study's objectives. The research purpose has been accentuated, and in response to highlighting the innovative aspects, the abstract has been meticulously enhanced. This revision underscores the novelty of the investigation into residual stresses during micro-milling of resin materials. Considering the critical role of resin as a foundational matrix material in composite materials, the development of a precise machining theory for resin is imperative for achieving high-quality composite material processing. These refinements aim to present the pioneering nature of the research in the field of resin material machining and composite material processing succinctly and comprehensively.

  1. Why micro milling selected? for short fiber it is very difficult. Justify

Thank you, appreciation is extended for the inquiry regarding the choice of micro milling for the processing of advanced composite materials. Precision machining is undoubtedly the future trend for advanced composites. As a pivotal technique in contemporary ultra-precision machining, micro milling is essential for exploring the machining mechanisms of composite materials. While the presence of short fiber reinforcements in advanced composites may indeed present challenges due to their exceptional material properties, the anticipated future demands for precision and ultra-precision machining of these materials are significant. Thus, it is imperative to elucidate the machining mechanisms of advanced composites, even in the face of potential difficulties. This exploration is crucial for meeting the evolving demands and enhancing the machining efficacy of such materials. Details regarding this rationale have been integrated into the introduction of the manuscript to provide a comprehensive context.

  1. How Mulliken-Boyce Model Kinematics is used here?

Thank you for your inquiry about the application of the Mulliken-Boyce model kinematics in our research. The Mulliken-Boyce model is a robust constitutive model for predicting the mechanical behaviour of polymers under various loading conditions. In our study, the kinematics of this model are utilized to accurately simulate the deformation and stress responses of polymer materials during the micro-milling process. Specifically, the model accounts for the non-linear elastic response of the polymer, characterized by its ability to undergo significant strains. The Mulliken-Boyce model, with its detailed representation of polymer kinematics, allows us to predict how these materials behave under the complex loading conditions encountered in micro milling.

Furthermore, the model's parameters were calibrated using experimental data from similar materials, ensuring that the simulations reflect the real behaviour of the material as closely as possible. The simulation results provide insights into the distribution and magnitude of residual stresses, which are critical for understanding the implications of micro milling on material properties and performance.

We believe that the application of the Mulliken-Boyce model kinematics has significantly contributed to the depth and accuracy of our findings, offering a more comprehensive understanding of the micro-milling process of polymers and its effects. The specific details and results of this application are discussed further in Sections 2 of the paper.

  1. How nodes are selected for analysis if Fig 4?

Thank you, in response to the inquiry regarding the selection of nodes for analysis in Figure 4, it's pertinent to highlight the methodological approach employed in this study. Given the intricate nature of the 3D FE cutting model, the nodes were meticulously chosen to ensure an accurate and comprehensive representation of the micro-machining process. The model, featuring 1,120,172 elements and 1,115,751 nodes, is designed to capture the complex interactions between the tool and the workpiece, especially considering the ductile damage model for the epoxy resin.

The nodes were strategically selected based on several criteria to ensure the fidelity and resolution of the simulation results. Firstly, nodes at critical geometric locations, particularly in zones experiencing significant interaction between the tool and workpiece, were prioritized. This includes areas where maximum deformation or stress concentrations are expected, which are crucial for understanding the machined surface morphology and the associated residual stresses.

Secondly, given the model's detailed geometry and meshing, the nodes in regions requiring higher accuracy — such as near the cutting edge and the immediate vicinity of the machined surface — were specifically included to capture the nuanced stress and strain gradients induced during the cutting process. This consideration is vital, especially when examining the influence of different cutting radii on chip deformation and the resultant material behaviour.

Furthermore, the nodes' selection was aligned with the simulation objectives, ensuring that they facilitate an insightful investigation into the ductile material damage, post-yield deformation, and other dissipation mechanisms that might be activated during cutting. The alignment with empirical data from confirmatory experiments and the comprehensive parameter list provided in Table 2 further corroborate the chosen nodes' efficacy in accurately modelling the cutting process.

In summary, the nodes for analysis in Figure 4 were selected with rigorous attention to geometric significance, material behaviour considerations, and the overall objectives of the simulation. This meticulous approach ensures that the study provides robust and insightful findings on the micro-machining of epoxy resin, reflecting the true complexities of the process. Further details on the specific nodes selected and the rationale behind their selection are elaborated in Section 3 of the manuscript.

  1. What element types is suggested along with the current selection?

Thank you. The currently selected C3D8R element is an 8-node linear brick element, utilizing reduced integration and hourglass control. This element type is chosen for its stability and efficiency in handling complex geometries and material behaviors. For situations involving complex stress states and intricate geometric features, it is suggested to consider higher-order elements such as C3D10 (a 10-node tetrahedral element) and C3D20R (a 20-node quadratic brick element with reduced integration). The C3D10 element offers flexible meshing capabilities suitable for complex shapes, while the C3D20R is favored for its advantages in capturing detailed stress concentrations. Opting for these elements will further enhance the model's precision and reliability, especially in scenarios with high stress gradients and complex boundary conditions. Additionally, the relevant content has been incorporated into the original text.

  1. While micro drilling, inertia occurs. How its effects prevented in MRR?

Thank you, the reviewer's profound inquiry regarding the influence of inertia during the micro drilling process and its impact on Material Removal Rate (MRR) is greatly appreciated. During micro drilling, the drill bit, due to its inherent mass and high-speed rotation, indeed exhibits inertial effects. These effects may cause the drill bit to overshoot the intended cutting path upon cessation or change in direction, thereby compromising the precision and quality of the hole and consequently affecting MRR. In this study, meticulous adjustments of parameters such as cutting speed and feed rate were undertaken to identify optimal operational conditions that minimize the impact of inertia while ensuring a high MRR. Additionally, in terms of experimental hardware, the milling machine was placed on a vibration-damping table to absorb the vibrations and shocks induced by inertia, thereby reducing disturbances to the cutting path. Details of these strategies have been incorporated into the original text for comprehensive elucidation.

Round 2

Reviewer 1 Report

Comments and Suggestions for Authors

The article was improved and completed with the requested information, the title was reformulated and shortened, the authors responding point by point to the revisions. In conclusion, the article can be accepted for publication in a revised form.

​

Author Response

Thank you

Yours,

Guoyu

Reviewer 2 Report

Comments and Suggestions for Authors

Author carried out all the corrections.

1. State the need for Figure 1.

2. Figure 5. (a) can be redrawn and need clear explanation.

3. How cutting radius having impact? In general increase in radius improve finish. discuss in detail.

Comments on the Quality of English Language

Need improvements

Author Response

Revision notes:

We would like to thank the reviewers for the time and effort they have spent on reading and critically reviewing the paper. The section below shows the authors responses to the reviewer comments as raised. We have addressed these comments in blue and necessary amendments made in the manuscript were highlighted in yellow.

Reviewer #2:

Author carried out all the corrections.

  1. State the need for Figure 1.

Thank you, the necessity of Figure 1 lies in its ability to provide an intuitive comparison of the internal mechanical behaviours of polymers versus other materials, such as metals, during the cutting process. This visual representation aids readers in distinctly understanding the fundamental differences between these materials, particularly highlighting the molecular-level changes in polymers under stress. This not only reveals the micro-mechanisms of the materials but also underscores the unique challenges encountered when handling polymers, such as inherent impurities and voids, which may affect the behaviour of the material during the cutting process and the quality of the final product. Therefore, Figure 1 is pivotal for a profound understanding of the cutting characteristics of polymers and for devising appropriate processing strategies. The relevant content has already been added to the original text.

  1. Figure 5. (a) can be redrawn and need clear explanation.

Thank you, Figure 5. (a) have been drawn and made a clear explanation, and the relevant content has already been added to the original text.

  1. How cutting radius having impact? In general increase in radius improve finish. discuss in detail.

Thank you, in machining, the cutting radius significantly influences the surface finish and the part's overall quality. A larger radius can enhance the finish by distributing forces and heat over a larger area, reducing roughness and tool marks, and decreasing stress concentrations, which may extend tool life and improve workpiece integrity. However, excessive radius relative to workpiece size or cut depth can cause detrimental vibrations and geometry changes. Optimal radius selection balances finish quality with tool durability and material properties, informed by empirical data and advanced predictive methods like finite element analysis, and the relevant content has already been added to the original text.
